# Effectiveness of Clinic-Based Patient-Led Human Papillomavirus DNA Self-Sampling among HIV-Infected Women in Uganda

**DOI:** 10.3390/ijerph20166613

**Published:** 2023-08-20

**Authors:** Agnes Nyabigambo, Roy William Mayega, Themba Geoffrey Ginindza

**Affiliations:** 1Department of Public Health Medicine, School of Nursing and Public Health, University of KwaZulu-Natal, Durban 4000, South Africa; rmayega@musph.ac.ug (R.W.M.); ginindza@ukzn.ac.za (T.G.G.); 2Department of Community Health and Behavioural Sciences, School of Public Health, Makerere University, Kampala P.O. Box 7062, Uganda; 3Health Economics and HIV/AIDS Division (HEARD), University of KwaZulu-Natal, Durban 4000, South Africa; 4Department of Epidemiology and Biostatistics, School of Public Health, Makerere University, Kampala P.O. Box 7062, Uganda; 5Cancer and Infectious Diseases Epidemiology Research Unit, School of Nursing and Public Health, University of KwaZulu-Natal, Durban 4000, South Africa

**Keywords:** HPV, effectiveness, cervical cancer, HPV self-sampling, home-based, clinic-based, women living with HIV

## Abstract

In Uganda, the uptake of cervical cancer (CC) screening services is low, at 46.7%, among HIV-infected women, and only 9% of these women adhere to annual CC screening. Some studies have evaluated the possibility of community or home-based human papillomavirus (HPV) self-collected vaginal swabs, but not clinic-based HPV self-collected vaginal swabs. Therefore, we propose a study to determine the efficacy of clinic-based versus home-based HPV DNA self-sampling among HIV-infected women attending a rural HIV clinic in Uganda. We believe that a randomized, single-blinded trial would achieve this objective, and so we have chosen it to guide the study. Including a total of 382 participants from a rural HIV clinic, randomized into a ratio of 1:1 for clinic- and home-based HPV self-sampling, would allow us to appropriately ascertain the difference in the uptake of HPV self-sampling between the two arms. The Integrated Biorepository of H3 Africa Uganda Laboratory would be used as a reference laboratory for the HPV DNA extraction, typing, and sequencing. At baseline, modified Poisson regression models would be used to measure factors associated with the prevalence of HPV and uptake in both arms at baseline. Visual inspection under acetic acid (VIA), as a gold-standard test for CC to grade for CIN, would be performed at 0 and 6 months among a random sample of 75 women with a self-collected HPV sample. The difference in uptake could be determined using the intention-to-treat analysis. The difference in the groups by each variable would be summarized as the standardized mean difference (i.e., the mean difference divided by the pooled standard deviation). The predictors of the time for which participants would continue with HPV self-sampling in both arms, recovery, and Cox proportional hazards regression would be used. At the bivariate level, the associations between each independent variable and time, with the time of continuing HPV self-sampling, would be computed. Crude hazard ratios and their 95% confidence interval would be used in the presentation of the results, with *p*-values < 0.05 considered significant at the bivariate level. Incremental cost-effectiveness analysis (CEA) using a Markov model would be used to determine the cost of clinic-based HPV self-sampling. We believe that screening approaches to disease stratification could provide an insight into the merits and limitations of current approaches to the diagnosis of cervical cancer, and how these could eventually be implemented into HIV clinics in Uganda and other developing African countries. It is anticipated that the findings would guide the development of step-by-step guidelines for the HPV self-sampling approach.

## 1. Introduction

The highest burden of cervical cancer (CC) disease is mainly observed in Sub-Saharan Africa [1,2]. In Africa, 24.5% of cancers are caused by infections, as opposed to traditional risks such as smoking, alcohol, and an unhealthy diet. Human papillomavirus (HPV) contributes to 12.1% of all cancers in Africa, with the commonest being cervical cancer [3]. Cancer of the cervix is avoidable and treatable if detected early among women [4]. The disease is primarily preventable among non-sexually active girls aged 9–13 years through routine vaccination with the HPV vaccine [5]. It is also secondarily preventable among HPV-exposed women through screening (the Papanicolaou test (Pap smear), a visual inspection of the cervix with acetic acid (VIA) or Lugol’s iodine (VILI), and cervicography) to detect and treat precancerous lesions [6,7]. Precancerous lesions are usually treated through cryotherapy [8,9,10] or thermal ablation. 

Cervical cancer is the commonest cancer among Ugandan women. Annually, 3915 women in Uganda are afflicted by CC and, of these, 2160 die [11]. The main cause of CC among women is persistent cervical infection by HPV [11]. The prevalence of HPV infection among women in Uganda is one of the highest, at 33.6%. According to the HPV Information Centre, the uptake of screening is low in Uganda, which contributes to the high incidence rate of 47.5 per 100,000 per year.

Cervicography, Pap smears, and VIA are mainly performed by health providers via a colposcopy [12,13]. HPV testing is also performed via self-sampling approaches [14,15]. Self-sampling approaches can be implemented in the community (through campaigns) [16], or in the clinic [10]. However, the self-sampling approach at home requires samples to be transported to the laboratory for HPV testing. This approach poses questions around the attrition and viability of the sample, compared to samples collected using the self-sampling approach at the clinic. 

Uganda faces many barriers to provider-led CC screening, and these include health workers who do not know the importance of CC screening [17], inadequate awareness, and the low economic status of communities [18,19,20,21]. New CC screening approaches are evolving more in urban areas, compared to rural ones [17]. The VIA screening approach is primarily used by women in urban health centers, while little is known about CC screening in rural health facilities [20]. Innovations including male partner involvement [22], the use of pocket-sized colposcopes [23], leveraging into existing HIV care [23], and self-sampling by women [24,25,26,27] aim to further increase access to CC screening, and reduce the incidence of invasive CC in Uganda. 

HPV testing is globally recommended instead of Pap testing, because it is also conducted on patient-collected samples [18,28]. The HPV DNA self-sampling approach is usually conducted from the vagina, using a swab or cytobrush [29,30]. The HPV self-sampling approach has the potential to reduce the hindrances related to provider CC screening, and will support acceptability among women who could otherwise not participate [18]. In developed countries, HPV DNA self-sampling is 85% acceptable among female commercial sex workers [30]. It is observed that the prevalence of HPV in patient-collected samples is high compared with provider-collected samples [31]. There are mainly 15 HPV DNA subtypes that are directly related to CC disease, and these are types 16, 18, 58, 52, 66, 68, 51, 45, 73, 35, 59, 31, 53, 33, and 39. The most common HPV DNA genotypes are 16 and 18 [31]. 

Countries in SSA, such as Uganda, bear a high burden of the HIV/AIDS epidemic, and 50% of HIV-infected women barely access CC screening. Repeated HPV infection, and the cervix’s inability to shade off the HPV virus due to reduced immunity, occur mostly in HIV-positive women. As a result, they easily progress to pre-invasive cervical lesions [32]. In Uganda, only 46.7% of HIV-infected women are screened for CC. Additionally, only 9% of HIV-infected women adhere to the annual CC screening guidelines in Uganda [1,33]. Approximately 98.9% of Ugandan HIV-infected women do not find it necessary to screen for CC, and only 46% accept the self-sampling approach [34]. Barriers to women’s participation in provider-led CC screening commonly relate to the anxiety of finding the disease, distress around the virginal speculum, a lack of time, embarrassment about the procedure being conducted by a male health worker, the absence of the right doctor, and previous negative experiences [21].

This increased risk among HIV-infected women requires them to screen for CC annually. However, there is a paucity of studies that have determined the effectiveness of clinic-based versus home-based HPV self-sampling approaches among HIV-infected women. It is anticipated that HPV patient-led CC screening would be effective, and would increase the uptake of CC screening and, hence, the early detection and treatment of CC. Additionally, there is no clear policy or organized structure for a nationwide cervical cancer screening program in Uganda. Therefore, there is an opportunity to increase the interest among rural HIV-infected women in going for HPV testing, through the promotion of programs that involve self-sampling approaches to HPV sampling among women. This could contribute to a reduction in the incidence of invasive CC, and in the health and community consequences of the disease. 

There is a lack of studies determining the effectiveness of patient-led HPV screening at an HIV clinic, compared to home-based vaginal HPV self-sampling, among HIV-infected women. We predict that if HIV-infected women conducted self-sampling at the HIV clinic using a vaginal collection kit method that was reliable, efficient, and acceptable, it would reduce the waiting time to access CC screening from the HW, enhance the prompt receipt of results, reduce attrition, and further address barriers to CC screening in rural communities. The results from the study will direct us to the appropriate HPV self-sampling approach for disease stratification, which will be published, and used to design policy briefs. Additionally, the findings will also be used to develop health promotion programs related to the prevention of HPV and CC among women. 

## 2. The Health Promotion Model for the Clinic-Based HPV Self-Sampling Approach

The health promotion model (HPM) has been chosen as the conceptual framework to guide this study, because health-promoting behavior is interpreted as a broad conceptualization, including the secondary prevention of CC. The theory could be used to interpret the effectiveness of clinic-based HPV self-sampling, in comparison with home-based HPV self-sampling approaches. It could be used to assess the difference in the uptake, HPV prevalence, associated factors, performance of clinic-based screening among HIV-infected women and their experiences, and costs related to patient-led HPV screening. 

The HPM would be used to interpret the factors affecting the uptake of clinic-based versus home-based HPV self-sampling among HIV-infected women. It was chosen because it suits the tailoring of public health interventions and dissemination [35]. Public health experts, especially public health nurses, can play an important role in educating women through clinic-based strategies that suit our social and cultural setting [36]. As a result, this would contribute to the promotion of health among women living with HIV.

The three concepts that were chosen to guide this study were the concept of personal factors, the concept of situational influences, and the concept of health-promoting behavior. The concept of personal factors could be used to identify the demographic characteristics (age, parity, level of education, CD4 count, viral load, knowledge of CC screening, screening preference, ART status, WHO stage, STI status, ART adherence status, fear of finding disease, type of HPV 16/18 by sample), knowledge, sexual history, reproductive health history, social factors, economic factors, and culture, concerning the prevention of HPV and eventually cervical cancer. The concept of situational influences could support the identification of the personal, family, community (perceived embarrassment of the screening procedure, knowledge of the disease, family support, peer support groups), and health system factors (distance to the health facility, waiting time, appointment scheduling, counseling status, peer support group, trained HWs, referral, the return of laboratory tests), and the cost-effectiveness, of the clinic-based versus home-based HPV self-sampling approach.

Health-promoting behavior could be interpreted as activities related to HPV self-sampling, at the clinic or at home, among HIV-infected women. The model could also examine the relationship between personal factors and health-promoting behavior; situational influences and health-promoting behavior; and personal factors and health promotion. Additionally, the HPM has been widely used in health promotion research involving people of different age groups regarding participation in HPV self-sampling and CC screening [36]. Figure 1 below shows a synopsis of the relationships among the concepts that are examined in this research.

## 3. Research Implications 

### 3.1. Study Site

Geographical areas where women have multiple sexual partners are key risk factors for HIV and HPV transmission. Luwero district is found in the central region of Uganda, where 19% of women have multiple sexual partners, which is a key risk factor for HIV and HPV transmission (UDHS, 2016). This provides the basis used to justify the HIV clinic at Luweero Hospital, Luweero district (Figure 2) being considered as the study site. The clinic serves nearly 7000 people living with HIV/AIDS (PLWHA). Of these, 2557 are female patients. Approximately 750 women have not been screened at all, not been screened within the last three years, have abnormal screening results, or have been screened for >3 years with normal results. 

### 3.2. The Intervention 

As shown in Table 1 below, the HIV-infected women would be recruited prospectively, and randomized into a clinic-based or home-based self-sampling approach, to estimate the difference in uptake between the two groups. All women enrolled would collect a HPV self-sample, and the random sample of women would be followed, to estimate the difference in the continuation rates of uptake of HPV testing at 0 and 6 months.

### 3.3. The Study Population 

The study approach would be to enroll all women attending the HIV clinic aged 25–49 years who had never been screened, had not been screened within the last year (those who had screened for >1 year with normal results), or had abnormal screening results at the clinic, and to obtain their consent to participate.

### 3.4. The Phased Approach to Study Implementation 

We propose using a phased approach, to allow step-by-step learning, as we implement the study procedures to determine the effectiveness of the clinic-based HPV self-sampling approach. The three phases that would guide this study are detailed below. 

#### 3.4.1. Pilot Phase 

To understand the dynamics of testing the tools, randomization, and recruitment procedures, and the specimen collection, handling, transportation, and storage before shipment to the central laboratory, it is important to pilot the tools. A three-day pilot of the tools could be conducted at Mukono General Hospital, Mukono District, which is not part of the study. We would aim to pretest 30 questionnaires among 15 HIV-infected women per arm. Per arm, three women would undergo HPV self-sampling and a visual inspection of the cervix under acetic acid (VIA) at the clinic and would also participate in an IDI. The main purpose of the pilot phase would be to test the tools, randomization, and recruitment procedures, and the specimen collection, handling, transportation, and storage before shipment to the central laboratory. 

#### 3.4.2. Baseline Phase

At baseline, a cross-sectional study design would be used to determine the factors associated with the prevalence of, and those associated with the uptake of, HPV testing in both arms. We could, further, estimate the differences in the viability of the samples between the clinic-based and control groups. 

#### 3.4.3. Effectiveness Phase 

A prospective randomized controlled single-blinded trial has been considered to estimate the efficacy of clinic-based versus home-based HPV self-sampling among HIV-infected women at the HIV clinic. As women came to the HIV clinic, they could prospectively be randomized to clinic-based or home-based HPV self-sampling. This would be a single-blinded randomized controlled trial, in which the study participants would not know the arm to which they had been assigned. After recruitment and consent, a randomization Excel list would be generated, and participants randomized to the two study arms in the ratio of 1:1. However, a random sample of HPV-positive women who took up HPV testing in both arms at baseline could also be subjected to VIA. The women would be followed up at 6 months, to measure their continued uptake rate of HPV self-sampling. The period of six months has been considered as is the time for their refill treatment for ART. As the study is tailored so that the findings could be used to design screening approaches that were public, feasible, and cost-effective, a modified societal perspective could be used to model the cost-effectiveness analysis. Non-medical costs would be obtained from the interviews that we conducted with the women, and from the secondary data analysis of the budgets. However, comparisons for the transport costs, personnel time, patient waiting time, and time spent in the facility would be considered. 

### 3.5. Recruitment

To obtain robust outcomes, all midwives involved in the study would be initially trained in the study procedures. All eligible women attending the HIV clinic would be educated on the different procedures for the collection of samples for HPV testing. Women could prospectively be recruited and randomized as they came to the HIV clinic. The midwives would obtain women’s consent to HPV testing and VIA screening. The women randomized to the clinic-based HPV self-sampling group who had consented to and accepted the screening would be provided with an HPV sample collection kit to collect the vaginal swab and return it to the laboratory for HPV testing. The women randomized to home-based HPV self-sampling, who had consented to, and accepted, the screening, would receive the HPV testing kit from the community linkage personnel (CLP). The CLP could collect all the collected samples from the women and transport them to the laboratory at Luweero Hospital. As recommended by the World Health Organization (WHO,) all women who took up screening in each group would need to first undergo a pregnancy test called human chorionic gonadotropin (HCG). 

The randomly selected sample of 150 HPV-positive women who took up HPV testing in both arms would further be subjected to a VIA, which is accepted as a gold-standard test for grading CIN in low-income settings such as Uganda. All women with suspected CIN2 and CIN3 could be referred to Uganda Cancer Institute for further management. The study would leverage the existing referral systems for cancer patients. UCI is the only national center for cancer treatment in Uganda and, therefore all identified cases are referred to that center as a norm; hence, there is no need for a memorandum of understanding. This is as shown in Figure 3. 

### 3.6. Sample Size 

It has been estimated that a total of 382 HIV-infected women would determine the effectiveness of the clinic-based HPV sampling approach. In each arm, i.e., the clinic-based and the home-based HPV self-sampling, there would be 191 participants. The required powered sample for this sub-study has been determined based on the formula for comparative studies where the outcome is categorical (or a proportion) (Schlesselman, 1982). 

To answer the question of the facilitators and barriers to HPV screening in the rural HIV clinic, we would conduct 24 in-depth interviews (IDIs) among women aged 25–35 years and 36–49 years, including women who would decline CC screening, with or without a history of STIs, as indicated in Table 2, below.

The phenomenology design would be used to explore the deeper facilitators and barriers toward women regarding self-sampling, either at home or in clinical settings, in Luweero District Hospital, Uganda. The in-depth interview (IDI) guide would be translated from English to Luganda. Qualitative data analysis would be guided by content analysis techniques. The transcripts would be coded in NVivo 20.7.0. The coded text would then be used to generate categories of analytically meaningful data that guided the formation of themes, the interpretation of results, and the final write-up.

### 3.7. Sampling Procedures 

An electronic sampling frame of women aged from 25 to 49 years would be obtained via the Luweero district hospital HIV clinic. The women’s history of CC screening could be reviewed in their files. All women who had not been screened at all, not been screened within the last three years, had abnormal screening results, or had been screened more than 1 year ago with normal results, attending the HIV clinic, could be included in the sampling frame. The first participant would be randomly selected, and the subsequent women would be systematically selected using an interval of 2. Purposive sampling would be used to allocate women for IDIs, and expert opinions would be used to collect cost data from key informants and documents.

### 3.8. Sample Collection Procedures

Those who took up screening in both groups would be educated on the sampling procedure, and given a Qvintip kit to collect their samples. The Qvintip dry-based sample collection kit can be transported at room temperature, and is, therefore, appropriate to use in developing countries, as it does not require cold-chain transportation. Therefore, it would be appropriate to collect samples using the Qvintip kit, because would be is user-friendly in our settings. The samples stored at the HIV clinic lab would be transported to the Integrated Biorepository of H3 Africa Uganda as a reference laboratory for the HPV DNA extraction, PCR amplification, testing, and genotyping. 

### 3.9. Data Analysis

To determine the factors associated with the uptake, and HPV prevalence at baseline (0 months) in clinic-based versus home-based HPV self-sampling from the baseline data, descriptive statistics, as well as univariate, bivariate, and multivariate logistic regression would be considered. The difference in uptake over the study period (effectiveness of clinic-based versus home-based HPV self-sampling) could be determined using the intention-to-treat analysis. We could further determine predictors of the time to continue with HPV self-sampling in both arms, and recovery, and Cox proportional hazards would be used. Content analysis could be used to analyze the qualitative data regarding the facilitators and barriers to HPV self-sampling. To estimate the cost-effectiveness of the clinic-based HPV self-sampling approach, an Excel model could be built and validated to check for the logical flow of events. 

### 3.10. Ethical Considerations 

Ethical approval was obtained from the University of KwaZulu-Natal Biomedical Research Ethics Committee (BREC), Makerere University, School of Public Health Higher Degrees Research and Ethics Committee (MaKSPH-HDREC), and the Uganda National Council of Science and Technology (UNCST). The UNSCST has guidelines that support primary data collection, and the COVID-19 risk management plan for this project has been developed within the required guidelines. 

## 4. Conclusions

HPV self-sampling at the HIV clinic could be effective, and would increase the number of women using the service. The prospective implementation of the study will provide insights into the merits and disadvantages of the HPV self-sampling approaches for disease stratification, hence contributing to the best service design for Sub-Saharan Africa. The findings will also be used to guide a step-by-step process for providing HPV self-sampling services at HIV clinics. 

## Figures and Tables

**Figure 1 ijerph-20-06613-f001:**
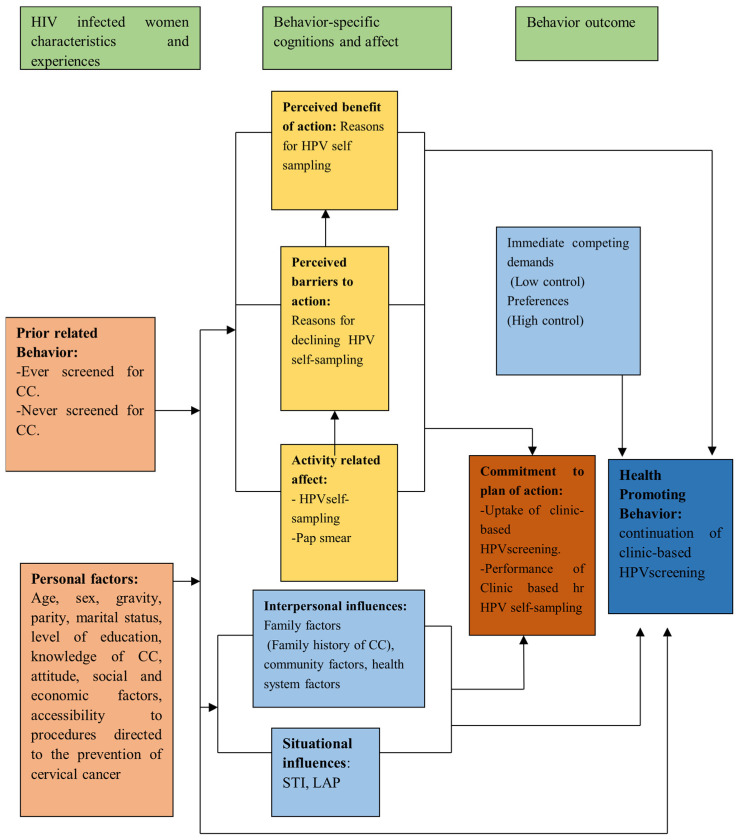
The synopsis of the Health Promotion Model for the study.

**Figure 2 ijerph-20-06613-f002:**
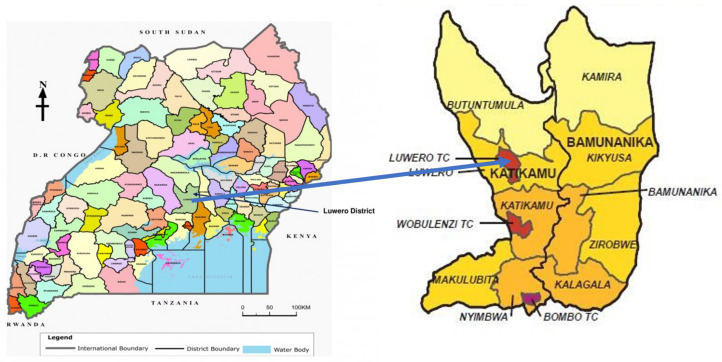
The map shows the location of Luweero district and Luweero district hospital.

**Figure 3 ijerph-20-06613-f003:**
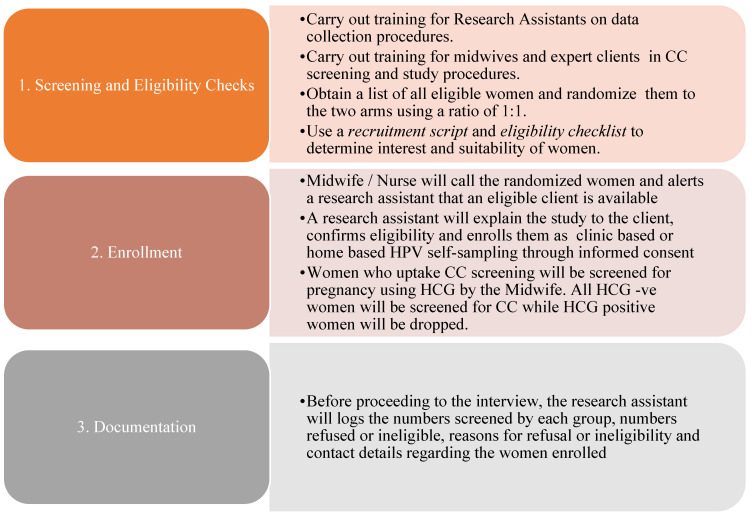
Process of recruiting women into the study.

**Table 1 ijerph-20-06613-t001:** Description of the clinic-based (intervention) group and home-based (control) group.

Clinic-Based (Intervention) Group	Home-Based (Control) Group
The intervention group will receive health education, and education on sample collection, and the midwife will be present to coach/mentor the women during the HPV self-sample collection. The woman will self-collect the sample and will also receive a visual inspection under acetic acid (VIA), which is a conventional CC screening approach at the clinic. The women will then take the HPV sample to the laboratory for storage at the facility and will also receive reminder calls or text messages for their next screening appointment. Women in the intervention arm will often receive call or text message reminders to come to collect the next testing kit from the midwife at the clinic.	The control group will receive a testing kit from the community linkages person (CLP)representative in the community who will also educate the women on sample collection. The information the CLP will give will be on sample collection only. The CLP will wait and take sample batches to the clinic for storage and, later, shipment to the laboratory. The women will also consent to come to the clinic to receive VIA services within a week.

**Table 2 ijerph-20-06613-t002:** Sample size estimation for IDI.

Age Category	>5–35 Years				36–49 Years				Declined CC Screening			
Sample collection preference	Clinic-based HPV self-sampling		Home-based HPV self-sampling		Clinic-based HPV self-sampling		Home-based HPV self-sampling		30–49 years		0–65 years	
Sample size	2	2	2	2	2	2	2	2	2	2	2	2

## Data Availability

The data from this study are the property of the University of KwaZulu-Natal (UKZN) and Makerere University. All interested readers are required to request the data from Makerere University, School of Public Health through the Principal Investigator, Agnes Nyabigambo. Email: anyabigambo@musph.ac.ug. Mobile phone: +256774135496.

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
