# Peer review of "Effectiveness of Clinic-Based Patient-Led Human Papillomavirus DNA Self-Sampling among HIV-Infected Women in Uganda"

_ijerph, 2023, doi:10.3390/ijerph20166613_

Round 1

Reviewer 1 Report (Previous Reviewer 1)

Please see the file.

Author Response

Dear Reviewer, 

Thank you for the comments. Below are the responses to the comments; 

  1. It is incorrect to delete information concerning the features of SS screening in Uganda, since this study is aimed specifically at studying the problems of screening and HPV self-sampling amongst Ugandan women (Lines 90-101).                                                                                                              Response: Lines 90-101 have been removed. 
  2. Table 1. should be left, because it explains the specifics between the clinic-based (intervention) group and home-based (control) group well. Response: Table 1 has been removed. 
  3. The same applies to Figure 3.                                                                Response: Figure 3 has been removed. 

Regards,

Agnes Nyabigambo (Correspondence Author) 

Reviewer 2 Report (Previous Reviewer 2)

My review comments have been answered/revised. I have no further comments

Author Response

Thank you!

Reviewer 3 Report (Previous Reviewer 4)

The edits that were made by the authors contributed to an overall improvement in clarity and conciseness; however, the main issue with this concept paper is that it will not generate a whole lot of interest for readers. See specific comments below:

1. I acknowledge the authors for their effort in the research design but the results would have a much greater impact. I would be more interested in seeing the effect of the different arms and measured factors on screening uptake. Included results from the pilot phase would even be better than no results. 

2. There are some typos and punctuation errors, such as missing periods throughout the text. Authors should carefully read through the entire document.

Author Response

Dear Reviewer

Below is the point to point response to comments raised; 

The edits that were made by the authors contributed to an overall improvement in clarity and conciseness; however, the main issue with this concept paper is that it will not generate a whole lot of interest for readers.

Response: Thank you.

See specific comments below:

  1. I acknowledge the authors for their effort in the research design but the results would have a much greater impact. I would be more interested in seeing the effect of the different arms and measured factors on screening uptake. Included results from the pilot phase would even be better than no results.                                                                                                   Response: It is a concept paper thus sharing results is not applicable.
  2. There are some typos and punctuation errors, such as missing periods throughout the text. Authors should carefully read through the entire document.                                                                                                  Response: Thank you, the typos and grammatical errors have been cleaned up in the document.                                                                                                                                    Sincerely,

         Agnes Nyabigambo ( corresponding author)

This manuscript is a resubmission of an earlier submission. The following is a list of the peer review reports and author responses from that submission.

Round 1

Author Response

Comment 1: The manuscript is dedicated to the important problem of HPV DNA self-sampling amongst women living with HIV to early cervical cancer detection. In particular, the authors address the problem of the effectiveness of clinic-based and home-based HPV DNA self-sampling among the female population in Uganda. Understanding the users’ preferences for self-sampling and the quality of sampling are critical for effective cervical cancer screening in this country.
Response 1: Thank you.

Comment 2: Undoubtedly, the results obtained during the study will be useful for healthcare workers and will help to improve national healthcare services.
Response 2: Thank you.

Comment 3: There are no serious flaws in the manuscript. The authors give a detailed justification for conducting this study.
Response 3: Thank you.

Comment 4: As to the Study’s Methodology, the authors have well thought out the research plan. The authors provide detailed tables that help to understand the algorithm of the planned research. The results that the authors plan to obtain in the course of the work are consistent with the presented material.
Response 4: Thank you.

Comment 5: However, the Concept Paper is of interest only to the supervisors of the project and without practical results it is no of great interest to the readers.
Response 5: The results will be published in a phased approach as indicated in the concept note.
Thank you once again for your kind consideration to review our manuscript.
Sincerely,

Reviewer 2 Report

This study protocol describes an intended randomized, single-blinded trial of clinic-based compared to home-based HPV self-sampling in Uganda.

I suggest clarifying a few things in the Methods

1)    Will recruitment of all women be performed at time of visit to the HIV-clinic? And then the clinic-based sampling will be performed immediately and the home-based sampling will be performed later in the village. Won´t this introduce selection-bias given that you will include only women that are inclined to go to the clinic?

2)    There is very little mentioned regarding the in-depth interview and how these will be analyzed.

3)    Only 6 months follow-up is quite short. Have you considered a longer follow-up?

Author Response

Comment 1: Will recruitment of all women be performed at the time of the visit to the HIV clinic? And then the clinic-based sampling will be performed immediately, and the home-based sampling will be performed later in the village. Won´t this introduce selection bias given that you will include only women that are inclined to go to the clinic?

Response 1: Computer-generated random numbers would be generated and allocated to women in each arm before they would be recruited. All study participants, health workers, and the research team would be blinded to the treatment arm, hence minimizing bias.

Comment 2: There is very little mentioned regarding the in-depth interview and how these will be analyzed.
Response 2: The phenomenology design would be used to explore the deeper facilitators and barriers of women regarding self-sampling either at home or in clinical settings in Luweero District Hospital, Uganda. The in-depth interview (IDI) guide would be translated from English to Luganda. Qualitative data analysis would be guided by content analysis techniques. The transcripts would be coded in NVivo 20.7.0. The coded text would then be used to generate categories of analytically meaningful data that guided the formation of themes, the interpretation of results, and the final write-up.

Comment 3: Only 6 months of follow-up is quite short. Have you considered a longer follow-up?

Response 3: The main outcome variable is the continuation rate of HPV-self-sampling at 6 months upon ART refill at either the HIV clinic or in the community.

Thank you once again for your kind consideration to review our manuscript.
Sincerely,

Reviewer 3 Report

Strengths

1.      This paper has the potential to address an imperative public health issue. Background information is credible and provides a solid foundation for the necessity of the study. Additionally, the explanation of the use of the theoretical framework was well placed and will be useful to the reader in further understanding the study. Figure 1 helping to visualize the theoretical framework was a definite strength.

2.      The phased approach for implementation seemed appropriate considering the logistics involved.

Suggestions for improvement

1.      There are a few areas in the introduction where information could be more concise or omitted. For instance, in paragraph 5 there is redundancy about the information that “repeated HPV infections” are the main contributor to cervical cancer, as it is stated earlier in paragraph 2. Additionally, the last 2 sentences of the introduction explaining what the results will help accomplish might be better place in the discussion/conclusion/implications.

2.      There are small typos in the article that would help the clarity of the study if they were corrected. (i.e., in the abstract, on page 3 HBM being HPM, etc.)

3.      Figure 3 is not referenced in the body text and may not be necessary for further explanation to the reader as it does not significantly contribute to further understanding. Overall, there are numerous figures/tables and while some are extremely helpful in visualization (Fig 1), others may be reviewed for importance in contribution to the article.

4.      There are a lot of components to this study and so clarifying the aims/outcomes of each area (randomized trial component, qualitative IDIs, cost-effectiveness, etc.)

Author Response

Comment 1: This paper has the potential to address an imperative public health issue. The background information is credible and provides a solid foundation for the necessity of the study. Additionally, the explanation of the use of the theoretical framework was well-placed and will be useful to the reader in further understanding the study. Figure 1 helping to visualize the theoretical framework was a definite strength.
Response 1: Thank you!

Comment 2: The phased approach for implementation seemed appropriate considering the logistics involved.
Response 2: Thank you!

Comment 3: There are a few areas in the introduction where information could be more concise or omitted. For instance, in paragraph 5 there is redundancy in the information that “repeated HPV infections” are the main contributor to cervical cancer, as it is stated earlier in paragraph 2. Additionally, the last 2 sentences of the introduction explaining what the results will help accomplish might be better placed in the discussion/conclusion/implications.
Response 3: Paragraph 5 of the introduction has been deleted. The last 2 sentences of the introduction explaining what the results will help accomplish have been placed in the discussion/conclusion/implications.

Comment 4: There are small typos in the article that would help the clarity of the study if they were corrected. (i.e., in the abstract, on page 3 HBM being HPM, etc.)
Response 4: The typo of HBM in the abstract has been replaced with HPM.

Comment 5: Figure 3 is not referenced in the body text and may not be necessary for further explanation to the reader as it does not significantly contribute to further understanding. Overall, there are numerous figures/tables and while some are extremely helpful in visualization (Fig 1), others may be reviewed for importance in contributing to the article.
Response 5: Figure 3 has been removed from the text body as it may not be necessary.

Comment 6: There are a lot of components to this study and so clarifying the aims/outcomes of each area (randomized trial component, qualitative IDIs, cost-effectiveness, etc.)
Response 6: The aims /outcomes of the randomized trial component, qualitative IDIs, and cost-effectiveness have been clarified. Thank you once again for your kind consideration to review our manuscript.
Sincerely,

Reviewer 4 Report

Increasing screening uptake is a topic that I am also interested in so I appreciated the proposal set forth in this concept paper. The overall design seems appropriate with a few things to consider:

1. It is stated that all enrolled women regardless of arms would be subject to VIA to grade CIN. Would this pose as a barrier since it involves healthcare provider involvement? Would this impact the results?

2. Although the authors claim that self-collected samples have higher prevalence of HPV compared to healthcare provider-collected samples, how will they know if this is also the case with their participants? Will a healthcare provider-collected specimen be compared to the self-collected specimen during the pilot phase?

3. The conclusion is rather vague and can be expanded upon to include thoughts on expected results and reasoning behind them.

4. There are several typos (misspelled words, punctuation, and spacing) found throughout the paper. Authors should take care in proofreading the entire paper for errors. 

Author Response

Comment 1: Increasing screening uptake is a topic that I am also interested in so I appreciated the proposal set forth in this concept paper. The overall design seems appropriate with a few things to consider:
Response 1: Thank you!

Comment 2: It is stated that all enrolled women regardless of arms would be subject to VIA to grade CIN. Would this pose as a barrier since it involves healthcare provider involvement? Would this impact the results?
Response 2: VIA will be performed after the woman has provided the HPV self-collected sample hence it would not impact on the results.

Comment 3: Although the authors claim that self-collected samples have higher prevalence of HPV compared to healthcare provider-collected samples, how will they know if this is also the case with their participants? Will a healthcare provider-collected specimen be compared to the self-collected specimen during the pilot phase?
Response 3: Our study measures HPV prevalence in self-collected samples only. We are not comparing HPV prevalence with healthcare provider-collected samples.

Comment 4: The conclusion is rather vague and can be expanded upon to include thoughts on expected results and the reasoning behind them.
Response 4: The conclusion has been expanded to include clinic guidelines that will support the HPV self-sampling approach at the HIV clinics.

Comment 5: There are several typos (misspelt words, punctuation, and spacing) found throughout the paper. Authors should take care in proofreading the entire paper for errors.
Response 5: All typos have been cleaned out of the paper. Thank you once again for your kind consideration to review our manuscript.
Sincerely,